# Assessment of Physicochemical Properties and Consumer Preferences of Multi-Millet Extruded Snacks Using a Fuzzy Logic Approach

**DOI:** 10.3390/foods14203517

**Published:** 2025-10-16

**Authors:** Tej Bhan Singh, Ramesh Kumar Saini, Ravinder Kaushik, Raju Sasikumar, Vivek Kambhampati, Seema Singh, Prince Chawla

**Affiliations:** 1Health Technology Cluster, School of Health Sciences and Technology, University of Petroleum and Energy Studies (UPES), Dehradun 248007, Uttrakhand, India; tejbhansingh1234@gmail.com (T.B.S.); rameshkumar.saini@ddn.upes.ac.in (R.K.S.); 2Department of Agribusiness Management and Food Technology, North-Eastern Hill University, Tura Campus, West Garo Hills, Chasingre 794002, Meghalaya, India; sashibiofoodster@gmail.com; 3Department of Food Process Engineering, National Institute of Technology, Rourkela 769008, Odisha, India; kambhampativ@nitrkl.ac.in; 4Department of Food Science and Technology, G. B. Pant University of Agriculture and Technology, Pantnagar 263145, Uttarakhand, India; seemasingh1040@gmail.com; 5Department of Food Technology and Nutrition, Lovely Professional University, Phagwara 144411, Punjab, India; princefoodtech@gmail.com

**Keywords:** extrusion processing, sensory evaluation, foxtail millet, snack formulation optimization

## Abstract

This study investigated the physicochemical characteristics, functional and technological properties, and consumer acceptability of multi-millet extruded snacks using a fuzzy logic approach, with particular emphasis on key sensory parameters: color, flavor, taste, and texture. Four formulations were developed using varying proportions of proso, kodo, and foxtail millets: S1 (50% proso millet), S2 (50% kodo millet), S3 (50% foxtail millet), and S4 (33% each). Physicochemical analysis indicated minimal nutrient (fiber, ash, and protein) loss during extrusion, while technological properties such as water absorption index (WAI: 5.71 g/g), water solubility index (WSI: 5.66–6.61%), and bulk density (0.13 to 0.16 g/cm^3^) yielded favorable results, contributing to improved texture and taste. The observed changes in starch structure positively influenced the organoleptic qualities of the snacks. Sensory evaluation, conducted by a 20-member panel, revealed that S3 (50% foxtail millet) received the highest acceptability scores, followed by S4, S2, and S1, and was rated as ‘very good.’ Among the sensory attributes, taste emerged as the most critical factor influencing consumer preference, followed by texture, flavor, and color. This study emphasizes the importance of integrating sensory analysis with fuzzy logic modeling to systematically optimize the formulation and processing conditions. This strategy enhances product quality by aligning technological functionality with sensory appeal, offering a robust framework for the development of consumer-preferred, health-oriented snacks and reducing the risk of market failure.

## 1. Introduction

In recent years, there has been a significant shift in human lifestyle toward healthier food choices, dietary supplements, and an overall focus on improving quality of life and well-being. Among these, millets and millet-based products such as snacks, puffed grains, and flakes have gained immense popularity due to their ability to boost immunity and reduce the risk of diseases. At the same time, the growing consumer preference for convenience has driven the demand for ready-to-eat (RTE) millet-based snacks due to their nutritional value. These snacks not only offer nutritional benefits but also come in a variety of shapes, sizes, and colors, catering to diverse consumer preferences. As a result, RTE millet-based snacks have become an integral part of modern diets, with their demand increasing significantly over the past two decades [1]. Currently, the snack market is flooded with starch-rich products made from corn, wheat, rice, potatoes, cassava, and sweet potatoes [2]. However, these are very poor in terms of protein, minerals, and fiber [3]. Furthermore, increased consumption of starch or starch-based products can cause several types of health disorders, such as obesity, hypertension, cancer, and elevated glycemic index [4]. Hence, food and nutrition scientists are actively exploring the best possible alternatives to traditional starch sources with optimum sensory attributes. The Indian diet is primarily cereal-based and rich in starch; however, challenges like population growth, reduced cultivable land, and climate change necessitate alternative solutions. Millets, known as “Shree Anna” or “Nutri-cereals,” are resilient, nutrient-rich crops with significant health benefits, making them suitable alternatives to starch-based snacks [5,6]. Millet-based extruded snacks offer a promising solution for the snack industry, promoting both nutrition and sustainability [7].

Traditionally, sensory evaluation relies on human perception; however, factors such as extraneous variables, modeling, complexities, and subjective judgment introduce uncertainty. Although methods such as the hedonic scale, paired comparison, duo-trio, and triangle tests are commonly used, they lack precision [8]. Traditional affective sensory evaluation is crucial for determining consumer preferences. In this study, a semi-trained sensory panel was employed, which proved suitable for evaluating the sensory characteristics of the pilot-scale developed product. However, the subjective nature of sensory data often necessitates computational approaches for accurate interpretation. Fuzzy logic has emerged as a powerful tool for modeling such data, offering a structured framework for analyzing consumer responses during product development [3,9]. To address this challenge, fuzzy logic evaluation provides a precise method for managing uncertainty and indistinctness in sensory analyses. This approach effectively analyzes both arbitrary and subjective data, allowing for significant conclusions on product acceptance, attribute strength, and ranking based on inputs from multiple sensory panelists [3]. Using this method, nutrient-dense multi-millet snacks were developed as healthier alternatives to traditional starch-based snacks. The extruded snacks were then evaluated through sensory analysis using the fuzzy logic approach, ensuring a more accurate and reliable assessment of their appeal.

## 2. Materials and Methods

### 2.1. Raw Materials and Blend Preparation

All millets, viz., kodo, proso, and foxtail (Farmers Grains Pvt. Ltd., Tiruchirappalli, Tamil Nadu, India) were procured and pulverized into flour (Chopin Laboratory CD-1 mill, Villeneuve-la-Garenne, France), and stored in a cool and dry place for further processing. Millet flour was mixed in different ratios to make blends for product formulation, as shown in Table 1.

### 2.2. Moisture Shifting

All the blends (Table 1) were mixed using a laboratory mixer (Plentary laboratory mixer, Spar SP-800), placed in an airtight pouch, and kept in a cool and dry place to avoid further moisture gain. Moisture content was analyzed according to the AOAC (1999). The initial moisture contents of S1, S2, S3, and S4 were 9.59, 9.44, 9.64, and 9.31%, respectively. The flour blends were conditioned to 20% moisture by adding water according to the formula (eq 1) suggested by [10]. After conditioning the flour blends, they were transferred into zip-lock pouches, and all the blends were equilibrated. Finally, the samples were stored in a refrigerator at 4 °C for 12 h.(1)$$Water\;need\;to\;be\;added\;=\;\left(\frac{100\;-\;intial\;moisture\;content\;of\;blends\;\left(\%\right)}{100\;-\;desired\;moisture\;content\;of\;blends}\right)\;\times \;Total\;amount\;of\;blends$$

### 2.3. Extrusion Processing

A laboratory-scale co-rotating twin-screw extruder (Jinan-Saibianuo Machinery Co. Pvt. Ltd., Shandong, China) with a power of 14.5 kW and extrusion capacity of 5–15 kg/h (Figure 1). The extruder dimensions were 2500 × 700 × 1500 mm, the screw length was 735 mm, the screw diameter was 30 mm, and the rotational screw speed was 0–500 rpm. The extruder parameters were as follows: screw speed, 20 Hz; feed rate, 10 Hz; and temperature, which was divided into four zones: feeding, mixing, cooking, and die/exit zones, at 50, 70, 90, and 120 °C, respectively. The optimized temperature was taken from our previous study where we optimized the temperature [11]. The final extruded product was collected, dried at 40 °C for 8 h, and stored in airtight pouches for further sensory analysis.

### 2.4. Physicochemical Properties

Proximate analysis was performed, which included moisture (d.b.), fat, protein, and crude fiber. Carbohydrate content was estimated using the subtraction method. The values for moisture (d.b.), fat, protein, and crude fiber were subtracted from 100.

### 2.5. Moisture

Moisture content was estimated according to AOAC 2005 [12]. A powder sample (2 g) was transferred to a pre-weighed petri dish. Subsequently, the petri dish was placed in a hot air oven at 105 ± 1 °C for 4 h until a constant weight was achieved.$$Moisture\left(\%\right)=\frac{Initial-final\;weight}{weight\;of\;sample}\;\times \;100$$

### 2.6. Crude Fat

Crude fat was calculated using the AOAC 2005 [12] Soxhlet extraction apparatus. A sample (5 g) was placed in a thimble with 90 mL of petroleum ether in a Soxhlet beaker and heated at 85 °C until the solvent evaporated completely. The beaker was then placed in a hot air oven for drying and weighed.$$Crude\;fat\left(\%\right)\;=\;\frac{Wieight\;of\;beaker\;with\;oil\;-\;Weight\;of\;empty\;beaker}{weight\;of\;sample}\;\times \;100$$

### 2.7. Protein Estimation

The Kjeldahl method was used to estimate the total protein content [11]. A 4 g extruded sample was digested with potassium sulfate and copper sulfate (1:2) in concentrated H_2_SO_4_ for 3 h. After cooling, distillation was performed using NaOH and boric acid, and the released ammonia was titrated with 0.1 N HCl.Protein (%) = A1 − A2 × N × 14.007 × 6.25/weight of sample$$Nitrogen=\frac{A1\;-\;A2\;\times \;N\;\times \;14.007}{weight\;of\;sample}\;\times \;100$$Protein (%) = Nitrogen (%) × 6.25
where *A*1 is the volume of 0.1 N HCL consumed for titration, *A*2 is the volume of 0.1 N HCL, *N* is the Normality of HCL, 14.007 is the atomic weight of nitrogen, and 6.25 is the conversion factor (nitrogen to protein).

### 2.8. Crude Fiber

For fiber estimation, 1 g of fat-free dried sample was digested with 1.25% H_2_SO_4_ for 45 min, followed by 1.25% NaOH for 30 min. The residue was incinerated in a muffle furnace at 550 °C for 3 h, cooled in a desiccator, and then weighed [12].$$Crude\;fiber\left(\%\right)\;=\;\frac{Initial\;-\;Final}{weight\;of\;sample}\;\times \;100$$

### 2.9. Total Ash

The ash content in the samples was estimated using the standard method of AOAC (2005) [12]. A 5 g sample was placed in a pre-weighed crucible, charred on a heating mantle, and incinerated in a muffle furnace at 550 °C for 5 h. After cooling in a desiccator, the crucible was weighed to determine the ash content.$$Ash\left(\%\right)\;=\;\frac{Initial\;weight-Final\;weight}{weight\;of\;sample}\;\times \;100$$

### 2.10. Total Carbohydrate

The carbohydrate content of the sample was calculated using the subtraction method using the following equation [12]:$$Total\;carbohydarte:\;100\;-\;\{Protein\;\left(\%\right)\;+\;Fat\;\left(\%\right)\;+\;Moisture\;\left(\%\right)\;+\;Crude\;fibre\;\left(\%\right)\;+\;Ash\;(\%)\}$$

### 2.11. Technological Properties

The water solubility index (WSI) and water absorption index (WAI) were calculated using the following equations: The method suggested by [11] about 1 g sample is taken into centrifuge tube and centrifuged (CPR 24 plus Remi, Mumbai, India) at 3000 rpm for 10 min, and then, supernatant was collected and dried. The difference in weights of the initial and final samples is taken as WAI, which is calculated using the following equation adopted from [13].$$WAI\left(\frac{g}{g}\right)\;=\;\frac{Weight\;of\;wet\;sediment}{weight\;of\;sample}$$$$WSI\;(\%)\;=\;\frac{Weight\;of\;dried\;supernatant}{weight\;of\;dried\;sample}\;\times \;100$$

### 2.12. Bulk Density

The bulk density was derived using the following equation method taken from [11]:$$Bulk\;density\;=\;\frac{W}{V}$$
where Wis the weight of the extrudate filled in the known volumetric cylinder, and Vis the volume occupied by the extrudate samples.

### 2.13. Color Analysis

The color values of the extrudates were measured using a Hunter Lab colorimeter (Colour flex EZ, Reston, Virginia) [13]. The grounded sample was placed into the vessel and made uniform to avoid irregular light scattering. The L, a*, and b* values of the extrudates were observed.

### 2.14. Sensory Analysis

The sensory evaluation of extruded snacks was performed by semi-trained 20 sensory panelists (25–30 years old), who were in good health, non-smokers, non-drinkers, non-allergic to millets, and interested in sensory analysis. Ethical approval was obtained before the sensory analysis (registration no. EC/NEW/INST/2022/2820). These individuals received basic training in recognizing and scoring key sensory attributes (color, flavor, taste, and texture) but were not trained to the extent of professional descriptive panels. The panelists were familiar with the sensory parameters of extruded snacks [14]. All the sensory attributes and nature of the product were explained to the panelists prior to the tasting, and they were informed about the sensory instructions that were mentioned in the scorecard.

Color: The appearance of the product, including its color in terms of brightness and browning after extrusionTexture: Crispness and crunchiness of the product during biteTaste: It is the impression of the product on individual perception. It includes sweet, salt, bitter, and off flavorsFlavor: Combination of taste and aroma during chewing.

Each attribute of each sample was rated as Not Satisfactory, Fair, Medium, Good, or Excellent.

The use of a semi-trained panel is intended to bridge the gap between analytical precision and consumer-like preference evaluation. During the extruded snacks sensory analysis, the panelists were guided to put ticks in columns with regard to fuzzy scale related quality characteristics of products, viz., color, flavor, taste, and texture [15]. After each sample analysis, the panelists rinsed their mouths with 0.5% saline water to eliminate the residue of the samples and then proceeded with further analysis [16].

### 2.15. Fuzzy Logic Implementation

Fuzzy logic analysis was employed to establish the relationship between sensory variables, linking independent factors such as color, texture, taste, and flavor with the dependent variables of acceptance and rejection [16]. A general flow chart (Figure 2) was used to study the fuzzy logic analysis of extruded snacks.

The extruded samples were ranked according to the triangular fuzzy membership distribution function. All the fuzzy scorecards of the extruded samples were translated into triplets and used to calculate the similarity scores of the extruded samples [16]. The sensory scale (0–100) was separated into five linguistic scale responses that represent NS (0 0 25), F (25 25 25), G (50, 25,25), VG (75, 25, 25), and E (100, 25, 0). Similarly, a 1–5 scale was also distributed for sensory attributes, where 1 = N.I. (0, 0, 25); 2 = SI (25, 25, 25); 3 = I (50, 25, 25); 4 = HI (75, 25, 25); and 5 = EI (100, 25, 0), respectively [8,17]. These values were considered for the evaluation of each judge’s responses toward each of the extruded snacks and sensory attributes.

The basic steps involved in the fuzzy logic model are as follows:The overall sensory scores of the extruded snacks in triplicate were determined.The membership function (MF) on a standard fuzzy scale was estimated.Computational analysis of the overall membership function on a standard fuzzy scale.Similarity values and rankings of the extruded snacks were estimated.Ranking of quality attributes of extruded snacks.

### 2.16. Triplets Associated with Sensory Scales

The triangular MF distribution of the sequence of sensory scores is demonstrated by three numbers, known as triplets. The triplets’ values linked to the triangular membership distribution function represent the five-point sensory scores, as shown in Figure 3. In Figure 3, Δabc demonstrates the function of membership distribution for the not satisfactory/not at all important segment, while Δac1d corresponds to the fair/somewhat important segment, etc. The first number in the triplet indicates the abscissa coordinate at which the MF is 1. The second and third numbers denote the left and right distances, respectively, from the initials, where the MF is 0.

### 2.17. Triplets for SS of the Extrudates and Quality Attributes

The SS of triplets for single sample was examined in terms of (i) the total of SS (ii) triplets connected with the sensory scale; and (iii) no. of panelists as represent by Equation (2) The overall sensory scores of each sample with regards of sensory attributes was calculated by using Equation (2)(2)$$SNX\;=\;\frac{n1(0\;0\;25)\;+\;n2\;(25\;25\;25)\;+\;n3\;(50\;25\;25)\;+\;n4\;(75\;25\;25)\;+\;n5\;(100\;25\;0)}{n1\;+\;n2\;+\;n3\;+\;n4\;+\;n5}$$
where S denotes the triplets for the sensory scores for sample n1, n2, n3, n4, and n5 represent the sensory score of judges with regards to corresponding linguistic terms viz. (n1 = NS, n2 = F, n3 = S, n4 = G, and n5 = E) related to corresponding triplets of sensory scale. Similarly, the quality attributes of the triplets of samples were calculated using Equation (2).

### 2.18. Triplet Representation of Relative Weights for Quality Attributes

The relative weights of the quality attribute triplets for the overall SS were determined using Equation (3).(3)$$QC(rel)\frac{QC}{Q(sum\;)}$$
where QCrel is represented as relative weights for color in triplets, QC; quality coefficient of an individual sample, Qsum; is the total of triplets for each quality attribute. Similarly, the terms used for flavor (QArel), taste (QTrel), and mouthfeel (QMrel) were calculated.

### 2.19. Triplets for Overall Scores of the Samples

Overall sensory scores of S1 were calculated using the formula given belowSO = (S C × QCrel) +(S A × QArel) + (S QTrel) + (S CT × QTrel)(4)
where SO represents the overall SS of the samples. QC, QA, QT, and QT represent triplets of color, appearance, taste, and texture, respectively. Equation (5) was used for the triplet multiplication for instances, triplet (m n o) with triplets (p q r).(m n o) × (p q r) = (m × p n × q + p × n m × r + p × o)(5)

### 2.20. Estimation of the Membership Function for the Standard Fuzzy Scale

The triangular spreading pattern on the standard fuzzy scale (6 scale), which is represented by F1, F2, F3, F4, F5, and F6, demonstrates the sensory score shown in Figure 4. The linguistic terminology for sensory scores was suitable for the analysis. The sensory scores for each MF followed a triangular distribution pattern. In a triangular distribution, the maximum value is 1. The values of the MF of F1 through F6 are defined by a set of 10 numbers.
**Membership Function****Sensory Response Articulation****Values of Membership**F1NS/NI1, 0.5, 0, 0, 0, 0, 0, 0, 0, 0F2F/SI0.5, 1, 1, 0.5, 0, 0, 0, 0, 0, 0F3S/I0, 0, 0.5, 1, 1, 0.5, 0, 0, 0, 0F4G/I0, 0, 0, 0, 0.5, 1, 1, 0.5, 0, 0F5VG/HI0, 0, 0, 0, 0, 0.5, 1, 1, 0.5, 0F6E/EI 0, 0, 0, 0, 0, 0, 0, 0, 0.5, 1

NS, not satisfactory; NI, not at all important; F, Fair; SI, somewhat important; S, Satisfactory; I, Important; G, Good; I, Important; VG, very good; HI, Highly Important; E, Excellent; EI, extremely important.

**Figure 4 foods-14-03517-f004:**
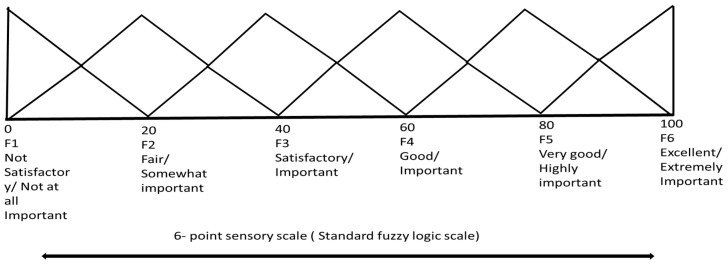
Demonstrated a standard 6-point sensory scale of fuzzy logic.

### 2.21. Calculation of the Aggregate Membership Function for Sensory Evaluation on a Standard Fuzzy Logic Scale

The MF of triplets (m n o) and all the triplets correlated with the overall SS calculated using the equation below. For triplets (m n o), the MF value is 1. The value of a function is 0 when the abscissa is less than a-b or greater than a + b. The function takes the value a for a given x-coordinate. Figure 5 shows the membership function of the triplets.Bx = (x − (a − b))/b for (a − b) < x < a(6)Bx = ((a + c) − x)/c for < a < x (a + c)= 0 for all other values of xB_x_ = 0 for x < (a − b) & x > (a + c)B_x_ = 1 at x = a(7)

### 2.22. Estimation of Similarity Values and Ranking of the Multi-Millet Extruded Snacks

The similarity of B values was calculated using Equation (8). The four samples were compared for their similarities. The sample obtained after calculating the similarity values was used to grade the sample quality. For example. If the similarity value was higher among all samples that were considered extremely important(8)$$S_{m}(F_{j},B_{x})\;=\;\frac{F_{j}\;\times \;B_{x}^{T}}{Max\;\;(F_{j}\;\times \;F_{j}^{T})and{(B}_{x}\;\times \;B_{x}^{T})}$$
where, Sm denotes a similarity value of the samples Bx; Fj represents F1, F2, F3, F4, F5; Bx is the MF value for sample x on the standard fuzzy scale; B_x_^T^ and F_j_^T^ are transposes of the matrix.

### 2.23. Estimation of Similarities Values According to Their Quality Attributes for All Samples

The same procedure was employed to obtain the quality characteristic ranking of the four extrudates, S1, S2, S3, and S4. MATLAB7.1 program (The MathWorks Inc., Natick, MA, USA) was utilized for the fuzzy logic evaluation of sensory data [15].

### 2.24. Statistical Analysis

All tests were performed in triplicate. Statistical analysis was performed using one-way ANOVA, means ± SD at a significance level (*p* < 0.05) using Origin Pro 2025 software.

## 3. Results and Discussion

### 3.1. Physicochemical Analysis

The major constituents of the multi-millet extrudates were analyzed. Extrusion processing had a non-significant (*p* > 0.05) impact on carbohydrate, fat, protein, fiber, and ash content, but a significant impact on moisture content was observed (Figure 6). The final moisture content of the extrudate ranged between 4 and 5.5%, marking an approximate 50% reduction owing to the thermal and mechanical effects during extrusion. This moisture loss primarily occurs through flash evaporation at the die exit, where high temperature and sudden pressure drop facilitate rapid water removal, simultaneously enabling product expansion and texture formation. Moisture plays a crucial role in the formulation of extruded snacks. The optimal moisture content for raw materials in extrusion processing ranges between 16 and 20%, as this ensures desirable expansion and improved texture in the final product. Fat content had no significant effect on extrusion, and protein, ash, and fiber levels showed minimal influence on process dynamics. Extrusion processing demonstrated no significant effect on total protein content, although partial denaturation occurred [18]. The carbohydrate content was statistically non-significant (*p* > 0.05) and remained largely unaffected (content-wise) by extrusion processing, with starch gelatinization—a structural transformation that enhances digestibility and texture without altering total carbohydrate composition. As per previous findings [6], the total carbohydrate content was slightly reduced after extrusion via thermomechanical impact on the starch content, which led to gelatinization, formation of starch—lipid complex, and conversion of sugar through Maillard reaction and caramelization. According to [18], there was a significant increase in the total carbohydrate content due to the thermal alteration of starch molecules, which led to cell rupture. The increase due to the dry weight basis calculation showed the impact of extrusion on carbohydrates. The calculation of carbohydrate (by the subtraction method) directly influences and shows an increase in carbohydrate content. The fat content of the extrudates was statistically reduced. This argument was supported by earlier studies [6,19], which reported that during the extrusion process, fat interacts with starch and forms a starch–lipid complex, which leads to reduced fat content. Total ash had no significant impact on extrusion, while crude fiber significantly decreased due to the thermal degradation of (β, 1–6) and conversion of insoluble into soluble fiber, which led to a reduction in total fiber [19,20].

### 3.2. Technological Properties

#### 3.2.1. WAI and WSI

The water absorption index was statistically significantly reduced (*p* < 0.05) through extrusion, and the WAI was estimated to be between 5.71 and 5.71 g/g (Figure 6). Currently, WAI refers to the starch gelatinization index. Under extrusion processing, the raw material experiences multiple forces like thermal, mechanical, and shear forces, which alter the native structure (chemical orientation) of the components [21]. Therefore, the water-binding sites in the existing structure were reduced, leading to a decrease in the WAI. Starch and protein are gelatinized and denatured, respectively. The chemical orientation of the gelatinized starch had a more open structure, which may have impacted the WAI [19,22]. Similarly, proteins and fibers change their structure under thermomechanical processes, which reduces their water-holding capacity. WAI is reduced by starch dextrinization, protein denaturation, and structural changes [20]. A lower water absorption Index (WAI) suggests that the starch has undergone extensive gelatinization, resulting in the breakdown of the granular structure and a more open, dispersed arrangement of starch molecules. This structural openness reduces the number of intact hydrophilic sites available for water binding. Consequently, such a matrix contributes to a desirable crispy and light texture in extruded snacks, allowing for rapid moisture loss and expansion during extrusion. In contrast, WSI increased significantly (*p* < 0.05) from 5.66 to 6.61% due to the breakdown of complex starch molecules into dextrin and smaller molecules. The T3 had the highest WSI (6.61%), while the lowest was found in T1 (5.66%). Higher fiber and protein contents create a barrier to starch gelatinization, which may lead to a lower degree of starch gelatinization, hampering the WSI [6,23]. Insoluble dietary fiber is hydrolyzed and releases soluble dietary fiber, which leads to an increase in the water solubility index of the extrudate. Structural changes under the extruder may result in the loss of molecular integrity, which causes changes in its structure. Starch gelatinizes, protein denatures, and unfolds from its native state, and fiber loses its structural orientation, leading to an increase in WSI.

The water absorption Index (WAI) and water solubility index (WSI) are influenced by starch modification, which plays a critical role in the development of snack texture. A higher WSI indicates greater expansion, which directly contributes to the enhanced crunchiness of the snacks. Similarly, a lower water absorption Index (WAI) indicates that the starch structure is more open and aligned. After complete gelatinization, starch possesses fewer water-binding sites, which significantly contributes to the desirable crunchy and soft texture of snacks. Both WAI and WSI depend on starch gelatinization, but they are negatively correlated with each other. Therefore, more active sites are present in the food sample after extrusion, which may lead to more water absorption, and WSI shows complete gelatinization. Complete gelatinization leads to the formation of soluble molecules that dissolve easily in water, enhancing the flavor release, texture uniformity, and overall palatability of the product. This indicates the cooking quality of starch under extrusion and its good quality characteristics [24].

#### 3.2.2. Bulk Density

Bulk density is a crucial physical parameter of extruded snacks from a commercial viewpoint, and density majorly influences packaging requirements [15]. The bulk density was between 0.13 and 0.16 g/cm^3^. No statistically significant difference (*p* > 0.05) was observed. Density is directly proportional to the expansion of the extrudates. More expansion, less density, and vice versa. Extrudates primarily composed of starch may significantly impact the density of the product via structural changes under high temperatures, leading to expansion. Complete gelatinization leads to good expansion, which dictates a less dense product [25]. The minimum density was found in T3 due to the presence of foxtail millet, which indicates more expansion and crunchiness, directly influencing the snack preference. Less foxtail density was observed in the study. Higher density was accompanied by a poor-quality preference by consumers. A similar study was reported by [23]. The density of a product, especially in extruded snacks, depicts the hardness of that product. If the density of a product is lower, its hardness is also lower. A lower bulk density was observed due to the flash-off effect, which occurs because of the sudden drop in pressure at the die exit during extrusion. This rapid pressure reduction caused a significant portion of the internal moisture to vaporize instantly into steam, leading to the expansion of the product structure and the formation of a porous texture [6,18]. Consequently, this expansion reduces the overall density of the extrudate and produces a desired expanded, crisp, and porous texture.

### 3.3. Color Analysis

Color is one of the most crucial parameters in terms of sensory attributes. The color of a product plays a crucial role in consumer perception, directly influencing its acceptance and rejection. In extrusion processing, color development is primarily driven by thermal reactions, such as the Maillard reaction, caramelization, browning, and pigment alteration, leading to color change. In this study, the L*, a*, and b* values of the blends ranged between 68 and 70, 3.17 and 4, 20.94 and 22.14, respectively, indicating light and yellowish hues in color. After extrusion, the L*, a*, and b* values ranged between 64.93 and 65.57, 4.12 and 4.32, and 24.49 and 24.87, respectively (Figure 7). The color of the extrudates was significantly impacted (*p* < 0.05), with a reduction in L* value and an increase in a* and b* values. The L* value decreased due to the combined effect of mechanical and shear forces under thermal conditions, causing the Maillard reaction between amino acids and reducing sugars. The L value indicates the lightness of the product [26]. The a* and b* values increased due to the thermal impact on the pigment properties and the formation of new compounds via caramelization and the Maillard reaction [27]. The lighter appearance of the extrudates can be attributed to an increase in the surface area after extrusion, which enhances light dispersion from the surface, resulting in a brighter or lighter visual appearance [24]. These color variations between the samples are sufficient to perceive differences that can be easily judged and observed by the panelists. When color differences are subtle, the semi-trained panel may easily shift their perception to more dominant attributes, such as taste or texture. This tendency of panelists may lead to bias in the sensory scoring process. However, the semi-trained panelists followed a standardized protocol to minimize such effects. Nevertheless, the color value of the extrudate in overall acceptability cannot be ruled out, and it is considered a limitation in the interpretation of the results.

### 3.4. Sensory Analysis

The semi-trained panelists provided valuable subjective feedback on product acceptability. Sensory analysis of millet-based extrudates was performed using standard fuzzy logic analysis. The sum of the SS was stated and judged by semi-trained panelists for different extrudates with different quality attributes in triplets, as shown in Table 2. The sensory scales are represented as NS, F, M, G, and Ex. In the given table, the observations based on panelist S3 were found to be good/excellent by the maximum number of judges’ preferences in terms of all quality attributes. The triplet related to the SS of multi-millet extruded snacks with different compositions is demonstrated in Table 2. For all samples, was calculated using Eq 1. Triplets of sensory grades and sensory scores were judged by the panelists. The S1C (47.50 25.00 23.75), S1 F (38.75 22.50 25.00), S1 T (47.50 21.25 22.50), and S1 T (40.00 20.00 23.75) are triplets for S1 in their color, flavor, taste, and texture attributes, respectively. The sensory scores were considered for general quality attributes, such as not important (NI), of some importance (SI), important (I), very important (VI), and extremely important (EI). The sum of the panelists with different judgments of sensory scores of multi-millet extruded snacks and their quality attributes is presented in Table 3. Among the 20 judges, 10 scored taste as extremely important, and taste was an important parameter for multi-millet extruded snacks from the consumer acceptability point of view among the four parameters. Furthermore, it was verified by optimizing the calculation of the similarity values of different extruded snacks in terms of quality attributes.

The overall sensory scores were estimated using Equation (3) for each sample. Table 1 demonstrates the triplets of the sensory scores, and their relative weightage was calculated using Equation (4) and is represented in Table 2. For instance, for sample 1, the expression is

(47.50 25.00 23.75). (0.185 0.075 0.079) +

(38.75 22.50 25.00). (0.256 0.088 0.062) +

(47.50 21.25 22.50). (0.264 0.088 0.057) +

(40.00 20.00 23.75). (0.295 0.088 0.044)

The triplets for other samples S2, S3, and S4 were as

S1: 52.50, 21.25, 22.50

S2: 72.50, 25.00, 17.50

S3: 75.00, 25.00, 16.25

S4: 83.75, 25.00, 12.50

### 3.5. Overall Membership Functions of Sensory Scores on Standard Fuzzy Scale

A six-point sensory scale, designated as F1, F2, F3, F4, F5, and F6, respectively, was utilized to determine the sensory scores, as previously described. The membership function values corresponding to the standard vague scale are provided in Equation (5). The overall membership function values of the sensory scores for the samples on the standard fuzzy scale, Bx, were calculated using the following Equation (5), as previously described. For example; the sensory scores for sample 1 are the overall sensory scores (52.50, 21.25, 22.50), i.e., a = 52.50, b = 21.85, and c = 22.50. By utilizing Equation (5), the value of the MF of the overall SS in the fuzzy scale at x = (0 to 100). The obtained values are as follows:

**Values****B1**0.09640.36980.64320.91661.00000.79790.50700.21610.00000.0000**B2**0.00000.18810.42330.65840.89361.00000.84730.56830.28930.0103**B3**0.00000.00000.13310.33320.53320.73330.93331.00000.80560.5141**B4**0.00000.14610.38450.62260.86061.00000.88250.59900.31550.0320

### 3.6. Similarity Values of Multi-Millet Extruded Snacks and Their Ranking

The similarity values and rankings of extruded snacks were analyzed, highlighting the comparative significance of various quality attributes that contribute to consumer acceptance or rejection [28]. The similarity values were calculated by combining the value of the membership function of the standard fuzzy scale and the overall membership function value of the sensory scores (Equation (6)).

For instance, the similarity values for S1 were:

0.0842 (Not satisfactory), 0.4549 (Fair), 0.7895 (Satisfactory), 0.5727 (Good), 0.1406 (Very good), 0.0000 (Excellent).

Similarly, the values for S2, S3, and S4are presented in Table 4. It was observed that sample 1 with the highest similarity value was satisfactory (0.7895), and S2 and S3 were good and very good, respectively. S4 was observed.

Based on the similarity values and sensory rankings in Table 4, the preference was as follows:**S3 > S4 > S2 > S1**

Therefore, S3 was found to be the best among the four samples, which constituted 50% foxtail millet, making it more soothing from a sensory perspective. S1 and S2 received the lowest preference due to their sensory attributes. S1 and S2, which contain higher proportions of proso and kodo, had maximum values, which resulted in imbalanced sensory attributes due to their high concentrations of fiber and other phytoconstituents. S4, with equal proportions of all three millets, showed moderate sensory scores, suggesting that an equal blend did not achieve optimal synergy with the sensory attributes observed in S3. A similar study was reported previously [29] in which multi-grain extruded flakes got ‘good’ in similarity values. Pasta was prepared from amaranth, oats, and rice, which was found satisfactory [30]. A study by Laghima Arora et al. [31] reported a variety of foxtail millet-based products, along with their sensory analysis. The sensory scores of the products showed a mean value of 8.00, indicating high product acceptability. This result highlights the importance of optimizing millet combinations to enhance consumer acceptance of extruded snack formulations. These findings underscore the importance of optimizing millet composition to enhance consumer acceptance of extruded snack formulations. The results of this study highlight the need for targeted formulation strategies to improve the textural and flavor profiles of extruded snacks, ensuring that the incorporation of different millet types aligns with consumer sensory preferences. Further research should focus on understanding the interactions between various millet compositions and their impact on sensory perception to develop nutritionally superior, widely accepted, and market-preferred snack products.

### 3.7. Quality Ranking of the Multi-Millet Extruded Food Snacks

The overall quality attributes of multi-millet extruded food snacks were observed to determine their sensory appeal. The quality of millet-based extrudates depends on their taste, texture, color, and flavor [32]. These quality attributes were determined, and similarities under different scale factors were calculated. The general ranking of the extrudates was computed based on these assessments. The overall membership function for the sensory score of quality attributes like color, flavor, taste, and texture, and the values for similarities F1, F2, F3, F4, F5, and F6 were calculated previously.

Table 5 depicts the similarity rankings for all quality attributes of the multi-millet extruded snacks. A comparative analysis of the similarity values revealed that ‘taste’ received the highest ranking among the four attributes. With a similarity value of 0.9300, “highly important” compared with color, flavor, and texture. Additionally, flavor and texture were also found under “highly important” with 0.8086 and 0.8523, respectively. Color was found under the “Important” category. Therefore, the order of these preferences was as follows:**Taste > Texture > flavor > color**

It was observed that all the factors were important for categorizing the multi-millet extrudates. The order of importance of these characteristic properties in determining the acceptability of other products varies significantly. This underscores the critical role of systematic and detailed studies, as conclusions regarding consumer preferences cannot be reliably drawn through mere observations. The highlights gained from such studies are invaluable in guiding the formulation and optimization of other extruded food snacks. By understanding the nuanced preferences of consumers, manufacturers can develop products that align more closely with market demands, thereby enhancing their potential for success. Future studies should include untrained consumer panels for comprehensive market-oriented validation.

## 4. Conclusions

In conclusion, multi-millets-based extruded snacks were successfully evaluated for sensory acceptability using a systematic fuzzy logic approach to rank and accept/reject the product. Among the four samples, Sample S3—comprising 50% foxtail millet, 25% proso millet, and 25% kodo millet—received the highest rating (‘Very Good’) and was ranked as S3 > S4 > S2 > S1. Taste emerged as the most important quality attribute, followed by texture, flavor, and color, emphasizing taste as a key parameter of consumer preference for multi-millet extrudates. The use of fuzzy logic added strength to the evaluation by effectively translating subjective sensory responses into objective, consistent rankings. The application of fuzzy logic enhanced the reliability of sensory evaluation by translating subjective responses into objective, consistent rankings. This minimized ambiguity in the panelists’ decisions and reduced the risk of product rejection in the market. Overall, the study demonstrates that millet-based extruded snacks are sensory acceptable and commercially promising, and that fuzzy logic serves as a valuable tool for supporting product development and market success.

Additionally, the formulations and process parameters were fixed, which may limit the generalizability of the findings across different production scales and ingredient variations. Future research should include a larger, more diverse consumer panel and explore a wider range of formulations and processing conditions to further validate and extend the applicability of these results.

## Figures and Tables

**Figure 1 foods-14-03517-f001:**
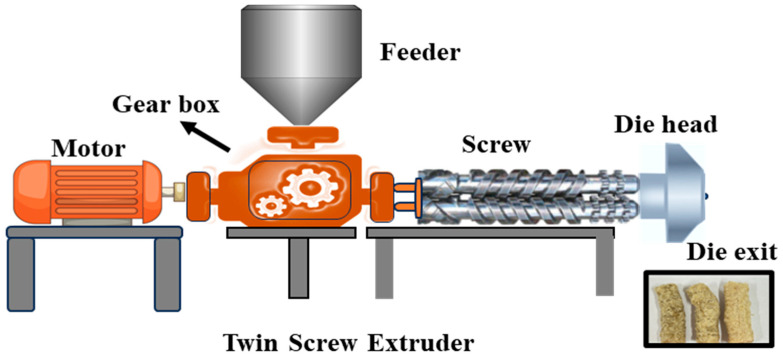
Components of the twin-screw co-rotating extruder.

**Figure 2 foods-14-03517-f002:**
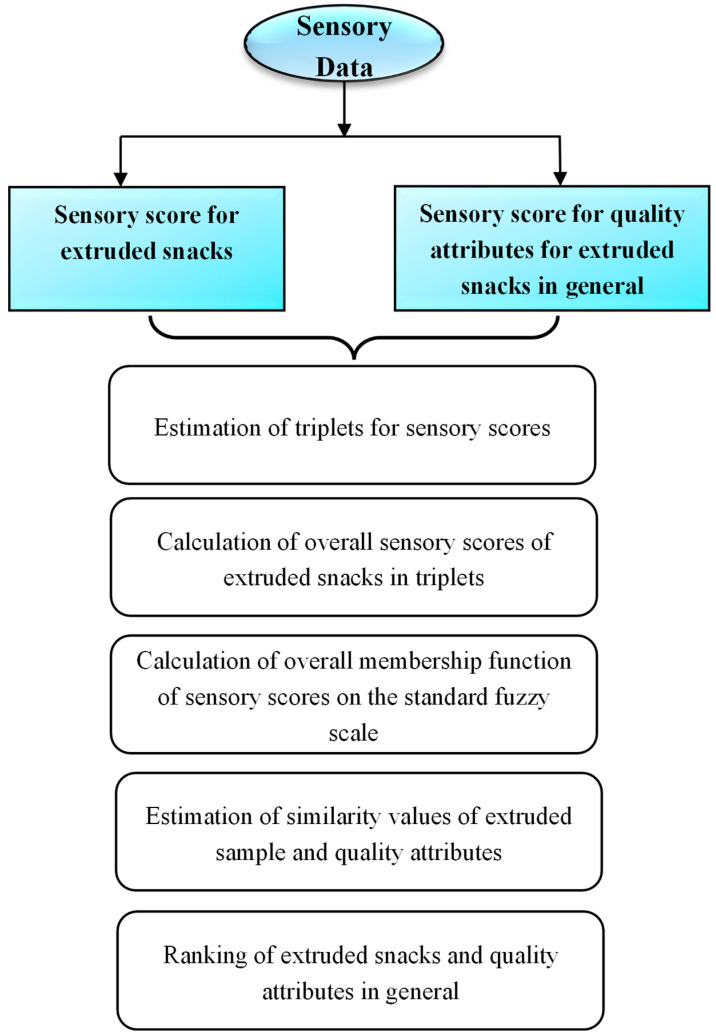
Process of sensory evaluation of extruded samples.

**Figure 3 foods-14-03517-f003:**
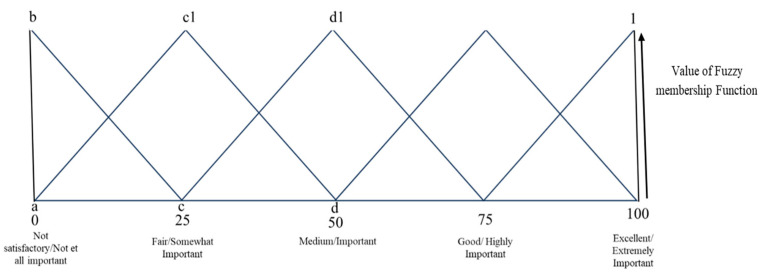
Represents a 5-point sensory scale with regard to triplets.

**Figure 5 foods-14-03517-f005:**
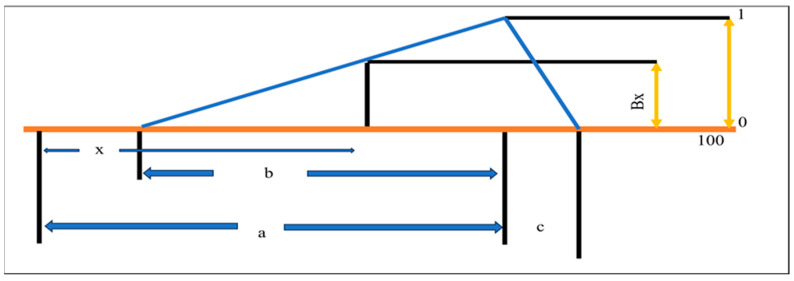
Demonstrating the triplets (a b c) into a membership function.

**Figure 6 foods-14-03517-f006:**
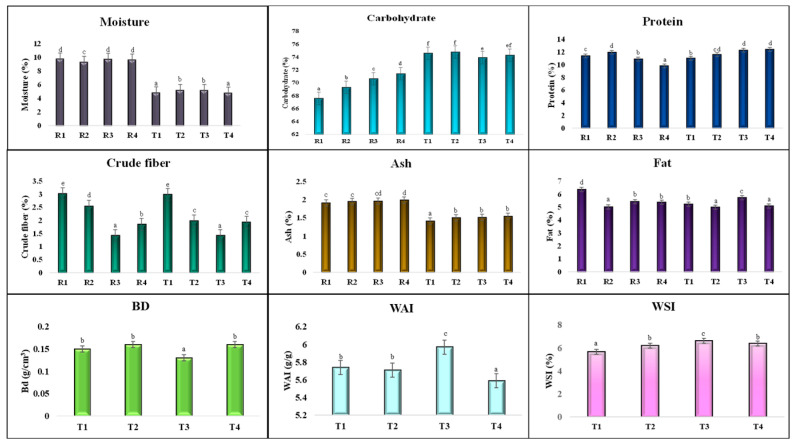
Physicochemical and functional properties of extrudates. All data are presented as the mean of n = 3, and statistical analyses and the superscripts denote the significant differences, with SD represented by error bars (*p* < 0.05). Where R1, R2, R3, and R4 are raw samples 1, 2, 3, and 4, respectively, and T1, T2, T3, and T4 are treated samples 1,2,3, and 4, respectively. BD-bulk density; WAI-water absorption index; WSI- water solubility index.

**Figure 7 foods-14-03517-f007:**
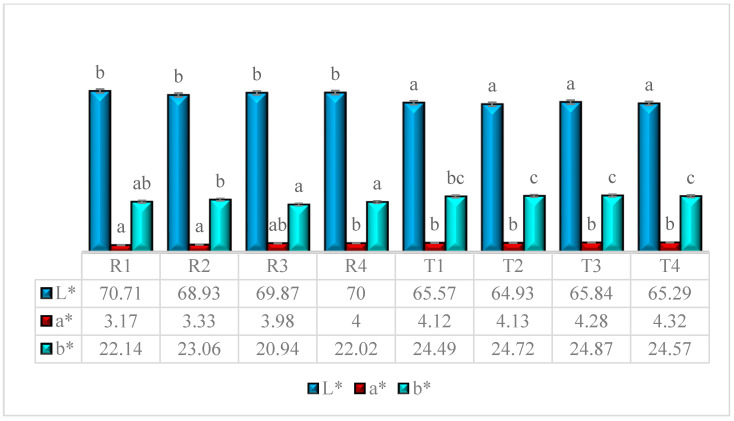
Color analysis of multi-millet extrudate. All values are represented as n = 3, and superscripts denote the significant differences within the group, with SD represented by error bars. where R1, R2, R3, and R4 is Raw samples 1, 2, 3, and 4, respectively, and T1, T2, T3, and T4, is Treated sample 1,2,3 and 4, respectively.

**Table 1 foods-14-03517-t001:** Different composition (%) for product formulation in different ratios.

S. No	Proso	Kodo	Foxtail
S1	50	25	25
S2	25	50	25
S3	25	250	50
S4	33	33	33

**Table 2 foods-14-03517-t002:** The Sum of SS judgment of multi-millet extrudates with different preferences and triplets was interlinked with sensory scores for their quality characteristics.

Quality Attributes	Sensory Scale Factor and Corresponding Numerical Values					
	Not Satisfactory	Fair	Medium	Good	Excellent	Triplets Related with Sensory Scores
**Color**						
**S1**	**0**	**10**	**3**	**6**	**1**	**47.50 25.00 23.75**
**S2**	**0**	**7**	**4**	**7**	**2**	**55.00 25.00 22.50**
**S3**	**0**	**1**	**4**	**8**	**7**	**76.25 25.00 16.25**
**S4**	**1**	**5**	**6**	**5**	**3**	**55.00 23.75 21.25**
**Flavor**						
**S1**	**2**	**10**	**3**	**5**	**0**	**38.75 22.50 25.00**
**S2**	**1**	**5**	**5**	**8**	**1**	**53.75 23.75 23.75**
**S3**	**0**	**2**	**5**	**6**	**7**	**72.50 25.00 16.25**
**S4**	**1**	**5**	**4**	**9**	**1**	**55.00 23.75 23.75**
**Texture**						
**S1**	**3**	**6**	**3**	**6**	**2**	**47.50 21.25 22.50**
**S2**	**1**	**5**	**3**	**7**	**4**	**60.00 23.75 20.00**
**S3**	**0**	**0**	**4**	**8**	**8**	**80.00 25.00 15.00**
**S4**	**4**	**2**	**6**	**5**	**3**	**51.25 20.00 21.25**
**Taste**						
**S1**	**4**	**6**	**5**	**4**	**1**	**40.00 20.00 23.75**
**S2**	**1**	**7**	**4**	**7**	**1**	**50.00 23.75 23.75**
**S3**	**0**	**4**	**5**	**5**	**6**	**66.25 25.00 17.50**
**S4**	**0**	**6**	**2**	**9**	**3**	**61.25 25.00 21.25**

**Table 3 foods-14-03517-t003:** Sum of the number of panelists.

Quality Attributes	NI	SI	I	HI	EI	Triplets for Sensory Grades	Relative Weights of Triplets (0-1)
**Colour**	3	3	5	7	2	QC = 52.50 21.25 22.50	QCrel = 0.185 0.075 0.079
**Flavor**	0	1	6	7	6	QF = 72.50 25.00 17.50	QFrel = 0.256 0.088 0.062
**Texture**	0	1	5	7	7	QT = 75.00 25.00 16.25	QTrel = 0.264 0.088 0.057
**Taste**	0	0	3	7	10	QT = 83.75 25.00 12.50	QTrel = 0.295 0.088 0.044

Where, QC, Sensory scores of color in triplets; QF, Sensory scores of flavor in triplets; QT, Sensory scores of taste in triplets; QCrel, relative weightage of color attributes in triplets; QFrel, relative weightage of flavor attributes in triplets; QTrel, relative weightage of texture attributes in triplets; QTrel, relative weightage of taste attributes in triplets.

**Table 4 foods-14-03517-t004:** Similarity Values of the multi-millet extrudates and their ranking.

Scale Factors	Sample 1	Sample 2	Sample 3	Sample 4
**NS (F1)**	0.0842	0.0263	0.0000	0.0207
**F (F2)**	0.4549	0.2634	0.0802	0.2381
**S (F3)**	**0.7895**	0.6339	0.3479	0.6153
**G (F4)**	0.5727	**0.7219**	0.6514	**0.7389**
**VG (F5)**	0.1406	0.3602	**0.6772**	0.3880
**E (F6)**	0.0000	0.0434	0.2455	0.0537
	**4th rank**	**3rd rank**	**1st rank**	**2nd rank**

Where NS is not satisfactory, F is Fair; S is Satisfactory; G is Good; VG is very good, E is Excellent, and F1 to F6 are its membership functions.

**Table 5 foods-14-03517-t005:** Similarity Values and ranking of quality attributes of the multi-millet extrudates.

Scale Factors	Color	Flavor	Texture	Taste
**NN (F1)**	0.0000	0.0000	0.0000	0.0000
**SN (F2)**	0.0824	0.0000	0.0000	0.0000
**N (F3)**	0.7177	0.1400	0.0800	0.0100
**I (F4)**	**0.8876**	0.7800	0.6800	0.3700
**HI (F5)**	0.2222	**0.8086**	**0.8523**	**0.9300**
**EI (F6)**	0.0000	0.1192	0.1851	0.4592
	**4th rank**	**3rd rank**	**2nd rank**	**1st rank**

Where NN—Not et al.l necessary; SN—somewhat necessary; N—Necessary; I—Important; HI—Highly important; EI—extremely important.

## Data Availability

The original contributions presented in this study are included in the article. Further inquiries can be directed to the corresponding author.

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
