# Peer review of "Assessment of Physicochemical Properties and Consumer Preferences of Multi-Millet Extruded Snacks Using a Fuzzy Logic Approach"

_foods, 2025, doi:10.3390/foods14203517_

Round 1
Reviewer 1 Report
Comments and Suggestions for Authors
Manuscript: foods-3695777-peer-review-v1
The manuscript entitled ``Assessment of physic-chemical properties and consumer preferences of multi-millet extruded snacks using a fuzzy logic approach´´ has been carefully reviewed. The submitted manuscript is adequately presented with relevant information in relation to sensory evaluation, however there are some issues that need to be more rigorously observed in order to be of a suitable standard for publication.
Comments
- Keywords: Authors should use words other than those present in the title in order to improve search engine indexing for the article.
- Abstract: The authors mention functional properties and these are rather technological properties of the product.
- Line 61: The mention or application of electronic nose has no justification within the introductory framework because it is not investigated. Therefore, the analysis between affective sensory evaluations and their comparison with fuzzi logic is more appropriate, as in “Comparative study on the hedonic and fuzzy logic based sensory analysis of formulated soup mix”. https://doi.org/10.1016/j.fufo.2022.100115
- Line 64: I consider that the mention of the authors is very forced to demonstrate a low importance of sensory evaluation when replaced by technological equipment, but this equipment cannot provide adequate information about tastes or preferences if first sensory equipment shows which product is more liked and then search with technology which are the key components that increase that preference or the other way around. I should not use such a determinant expression because there is still no efficient technology to replace the subjectivity of consumer judges representing a given social environment because a molecule that is liked in one region may be totally opposite to its preference in another region.
- Table 1: What does Mr. N° mean?
- Line 83-86: The information provided is the same as in table 1. Eliminate one of the two ways of explaining what the samples consist of.
- Line 96: Eq1 should be in italics at the side of the equation..
- Line 109: No mention of ash, which is part of the chemical composition of the food.
- 4: Is it necessary to develop the applied techniques that are so common? Perhaps mentioning the reference used and some specific specification is sufficient.
- Line 154: Ash determination is mentioned at one point in the methodology but no subtraction is made in the calculation by difference for carbohydrates.
- 11: They are rather technological or physico-chemical properties, but not so much functional ones.
- 14: It is not clear to me that trained judges evaluate tests as if they were untrained consumer judges to evaluate preference. Let's say this is a key concept in any food sensory evaluation book. I don't think it is correct that a person trained in discriminating attributes can have a subjective view. Although I have found other articles that apply a similar way of analysing, this leads to confusion in the data and incorrect interpretations of the results. In any case, if we wanted to measure attributes, we could work with a trained panel and, if we liked, with an untrained panel to combine the data and then process it. Now, if a new methodology is to be developed, mention should be made of this deviation in terms of a correct sensory technique that can subsequently give erroneous results, which is why in the introduction mention is made of the low accuracy of the sensory tests. The issue is the correct application of the right methodologies. The authors should make a more detailed explanation of this issue and why judges who have to define preference were trained and whether this does not imply a potential bias in the results.
- Line 182: How the authors have defined the sensory attributes tested as they do not mention what they are referring to when they talk about colour or texture, e.g.
- Line 245: The authors have forgotten to explain what QC is in the equation. Please add the definition.
- Line 372-373: These data sets correspond to the data of R and T?
- Line 501: Doesn't the lack of difference in colours between treatments cause trained judges to downplay its importance in preference assessment as they concentrate more on the indicators that are changing? Potential errors in the results obtained need to be taken into account.
Author Response
Reviewer 1: Comments and Responses
The manuscript entitled ``Assessment of physic-chemical properties and consumer preferences of multi-millet extruded snacks using a fuzzy logic approach´´ has been carefully reviewed. The submitted manuscript is adequately presented with relevant information in relation to sensory evaluation, however there are some issues that need to be more rigorously observed in order to be of a suitable standard for publication.
Reply: Thank you for taking the time to review our manuscript, ``Assessment of physic-chemical properties and consumer preferences of multi-millet extruded snacks using a fuzzy logic approach´´. We appreciate your valuable feedback and constructive criticism. We appreciate your comment and I accept, it enriching the quality of manuscript.
………………………………………………………………………………………………….
Comment 1: Keywords: Authors should use words other than those present in the title to improve search engine indexing for the article.
Response: Thank you for your feedback. The keywords have been revised and updated to: Extrusion processing; Sensory evaluation; Foxtail millet; Snack formulation; Optimization.
Comment 2. Abstract: The authors mention functional properties and these are rather technological properties of the product.
.
Response: Thank you for your observation. We agree with your comment and have revised the abstract accordingly by replacing "functional properties" with "technological properties" to accurately reflect the nature of the product characteristics discussed.………………………………………………………………………………………………….
Comments 3: Line 61: The mention or application of electronic nose has no justification within the introductory framework because it is not investigated. Therefore, the analysis between affective sensory evaluations and their comparison with fuzzi logic is more appropriate, as in “Comparative study on the hedonic and fuzzy logic based sensory analysis of formulated soup mix”. https://doi.org/10.1016/j.fufo.2022.100115
Response: Thank you for pointing this out. We agree that the mention of the electronic nose was not appropriate in the context of our study, as it was not part of our investigation. We have removed from line 61 and instead revised the discussion to focus on the comparison between affective sensory evaluations and fuzzy logic analysis, as suggested. The referenced article has also been cited appropriately to strengthen the context.
………………………………………………………………………………………………….
Comments 4: Line 64: I consider that the mention of the authors is very forced to demonstrate a low importance of sensory evaluation when replaced by technological equipment, but this equipment cannot provide adequate information about tastes or preferences if first sensory equipment shows which product is more liked and then search with technology which are the key components that increase that preference or the other way around. I should not use such a determinant expression because there is still no efficient technology to replace the subjectivity of consumer judges representing a given social environment because a molecule that is liked in one region may be totally opposite to its preference in another region.
Response: Thank you for your insightful comment. We acknowledge that the original statement may have overstated the limitations of sensory evaluation in comparison to technological tools. As you rightly pointed out, consumer sensory evaluation remains essential for understanding preferences within specific social and cultural contexts, which technology alone cannot fully capture. We have revised the sentence to reflect a more balanced perspective, emphasizing the complementary roles of sensory evaluation and technological analysis rather than suggesting one can replace the other.
We have also incorporated new references to strengthen the scientific basis of our data. These revisions enhance the clarity and accuracy of the reported information.
………………………………………………………………………………………………….
Comment 5: Table 1: What does Mr. N° mean?
Response: Thank you for your observation. I have made that correction in table 1.
…………………………………………………………………………………………………. Comment 6: Line 83-86: The information provided is the same as in table 1. Eliminate one of the two ways of explaining what the samples consist of.
Response: Thank you for your observation. We agree that the information was redundant. To improve clarity and avoid repetition, the textual description in lines 83–86 has been removed, as the sample compositions are already clearly presented in Table 1.
………………………………………………………………………………………………….
Comment 7: Line 96: Eq1 should be in italics at the side of the equation L105: again Table 1 reports
Response: Thank you for the suggestion. The formatting of all equation references (e.g., Eq. 1) has been updated throughout the manuscript to maintain consistency and adhere to standard formatting guidelines.
………………………………………………………………………………………………….
Comments 8: Line 109: No mention of ash, which is part of the chemical composition of the food.
Response: Thanks for your observation. We apologies for this mistake. We acknowledge the omission and have now included ash content in the description of the chemical composition to provide a more complete analysis of the food product.
………………………………………………………………………………………………….
Comment 9: Is it necessary to develop the applied techniques that are so common? Perhaps mentioning the reference used and some specific specification is sufficient.
Response: We agree that for commonly applied techniques, a brief description along with appropriate references and key specifications is sufficient. Accordingly, we have streamlined the methodology section by reducing excessive detail and retaining only essential information with proper citations.
………………………………………………………………………………………………….
Comment 10: Line 154: Ash determination is mentioned at one point in the methodology but no subtraction is made in the calculation by difference for carbohydrates.
Response: Thank you for pointing this out. We have revised the methodology section to clarify that ash content was included in the proximate analysis and appropriately accounted for in the calculation of carbohydrate content by difference.
…………………………………………………………………………………………………
Comment 11: They are rather technological or physico-chemical properties, but not so much functional ones.
Response: Thank you for your observation. We agree with your comment and have revised the terminology accordingly throughout the manuscript, replacing “functional properties” with “technological” or “physico-chemical properties” where appropriate to more accurately reflect the nature of the parameters discussed.
………………………………………………………………………………………………….
Comment 12: It is not clear to me that trained judges evaluate tests as if they were untrained consumer judges to evaluate preference. Let's say this is a key concept in any food sensory evaluation book. I don't think it is correct that a person trained in discriminating attributes can have a subjective view. Although I have found other articles that apply a similar way of analysing, this leads to confusion in the data and incorrect interpretations of the results. In any case, if we wanted to measure attributes, we could work with a trained panel and, if we liked, with an untrained panel to combine the data and then process it. Now, if a new methodology is to be developed, mention should be made of this deviation in terms of a correct sensory technique that can subsequently give erroneous results, which is why in the introduction mention is made of the low accuracy of the sensory tests. The issue is the correct application of the right methodologies. The authors should make a more detailed explanation of this issue and why judges who have to define preference were trained and whether this does not imply a potential bias in the results.
Response: Thank you for your detailed and insightful comment. We understand the concern regarding the distinction between trained and untrained panelists in sensory evaluation. We would like to clarify that in our study, Semi-trained panelists were used to evaluate the samples based on overall preference, aligning with standard affective sensory testing methods. We acknowledge the importance of using the correct methodology and have now clearly stated in the revised manuscript that the panel consisted of semi-trained individuals to avoid any confusion. Additionally, a brief explanation has been added to justify this choice and to acknowledge the potential limitations associated with subjective consumer-based evaluation.
………………………………………………………………………………………………….
Comment 13: Line 182: How the authors have defined the sensory attributes tested as they do not mention what they are referring to when they talk about colour or texture, e.g.
Response: Thank you for your comment. We acknowledge that the definitions of the sensory attributes were not clearly stated. In the revised manuscript, we have now included brief descriptions of each sensory attribute (such as colour, texture, taste, etc.) to clarify how they were defined and evaluated during the sensory analysis.
………………………………………………………………………………………………….
Comment 14: Line 245: The authors have forgotten to explain what QC is in the equation. Please add the definition.
Response: Thank you for your comment. We have now included the definition of QC (Quality coefficients of an individual sample) in the revised manuscript to ensure clarity in the equation presented on line 259.
………………………………………………………………………………………………….
Comment 15: Line 372-373: These data sets correspond to the data of R and T?
Response: Thank you for highlighting this point. We acknowledge the confusion and have revised lines 372–373 to clarify the origin and meaning of the data sets. The references to "R" and "T" were incorrect and have been removed or corrected to accurately reflect the actual variables used in the study.
………………………………………………………………………………………………….
Comment 16: Line 501: Doesn't the lack of difference in colours between treatments cause trained judges to downplay its importance in preference assessment as they concentrate more on the indicators that are changing? Potential errors in the results obtained need to be taken into account.
Response: Thank you for highlighting this point. We agree that minimal variation in colour between treatments could lead panelists—especially trained ones—to focus more on other distinguishing attributes, potentially influencing preference outcomes. While our panel consisted of semi-trained judges, this limitation has now been acknowledged in the discussion section, and the potential for such bias in interpretation has been noted as a consideration for future studies.
Reviewer 2 Report
Comments and Suggestions for Authors
1.All formulas need to be numbered separately.
2.How is the temperature range determined in the extrusion process?
3.The occurrence of abbreviations must be defined. For example, in section 2.12, BD? Please check the full text.
4.The statistical analysis method of the test needs to be described in detail in the material method.
5.Figure 6 requires a significance analysis.
6.Please explain why a lower water absorption index (WAI) indicates that the starch structure is more open and arranged? After complete gelatinization, starch has fewer water-binding sites, which significantly contributes to achieving the ideal crispy and soft texture in snacks?
7.Figure 7, significance letters between groups or within groups? This is difficult for readers to understand quickly. Please explain in detail, author.
8.Section3.5, (52.50, 21.25, 22.50), the numbers need to be separated by symbols.
9.The impact of this study on the food industry and the existing limitations need to be mentioned in the conclusion.
10.The readability of the article needs to be improved. Many writing parts do not strictly follow the requirements of the journal. Authors need to carefully check them. It is best to use professional language editors to carefully proofread the article.
Author Response
Reviewer 2
Comment 1: All formulas need to be numbered separately.
Response: Thank you for your valuable feedback on our review. We appreciate your recognition; The revised manuscript have been updated.
………………………………………………………………………………………………….
Comment 2: How is the temperature range determined in the extrusion process?
Response: Thank you for your question. The temperature range used in the extrusion process was optimized based on our previous study, which is now appropriately cited in the Materials and Methods section of the manuscript for clarity and reference.
………………………………………………………………………………………………….
Comments 3: The occurrence of abbreviations must be defined. For example, in section 2.12, BD? Please check the full text.
Response: We have carefully reviewed the manuscript and ensured that all abbreviations, including BD (Bulk Density) in section 2.12, are now properly defined at their first occurrence to enhance clarity and consistency throughout the text.
………………………………………………………………………………………………….
Comments 4: The statistical analysis method of the test needs to be described in detail in the material method.
Response: Thank you for your observation. We have revised the Materials and Methods section to include a detailed description of the statistical analysis methods used in the study, specifying the software, statistical tests, and significance levels applied to ensure clarity and reproducibility.
………………………………………………………………………………………………….
Comment 5: Figure 6 requires a significance analysis.
Response: We have now included a significance analysis in Figure 6, indicating the statistical differences among the samples using appropriate annotations with different letter along with a description in the figure caption for clarity.
………………………………………………………………………………………………….
Comment 6: Please explain why a lower water absorption index (WAI) indicates that the starch structure is more open and arranged? After complete gelatinization, starch has fewer water-binding sites, which significantly contributes to achieving the ideal crispy and soft texture in snacks?
Response: Thank you for your insightful comment. We have revised the explanation in the manuscript to clarify this point in the result and discussion section of WAI and WSI.
………………………………………………………………………………………………….
Comment 7: Figure 7, significance letters between groups or within groups? This is difficult for readers to understand quickly. Please explain in detail, author.
Response: Thank you for your insightful comment. We have revised in figure section. It is Within group.
………………………………………………………………………………………………….
Comment 8: Section3.5, (52.50, 21.25, 22.50), the numbers need to be separated by symbols Response: Thank you for your insightful comment. We have revised in that section.
………………………………………………………………………………………………….
Comment 9: The impact of this study on the food industry and the existing limitations need to be mentioned in the conclusion.
Response: Thank you for your valuable suggestion. We have revised the conclusion section to include the potential impact of this study on the food industry, particularly in the development of healthier, millet-based extruded snack products using fuzzy logic for sensory evaluation. Additionally, we have acknowledged the study’s limitations, such as the use of a semi-trained panel and the restricted number of formulations, which may influence the generalizability of the findings.
………………………………………………………………………………………………….
Comment 10: The readability of the article needs to be improved. Many writing parts do not strictly follow the requirements of the journal. Authors need to carefully check them. It is best to use professional language editors to carefully proofread the article.
Response: Thank you for your constructive feedback. We acknowledge the need for improved readability and adherence to journal guidelines. The manuscript has been thoroughly revised for language, grammar, and clarity. Additionally, professional proofreading has been undertaken to ensure that the writing meets the required academic and editorial standards.
All the file and changes are attached in the manuscripts.
Thanks for your valuable inputs, time, and efforts for reviewing it, which make the manuscript more scientific and authentic.
Round 2
Reviewer 1 Report
Comments and Suggestions for Authors
The authors have responded to the comments and the manuscript is now clearer and the article is accepted and can be published.
Reviewer 2 Report
Comments and Suggestions for Authors
None.